# Impact of Intercurrent Introduction of Steroids on Clinical Outcomes in Advanced Non-Small-Cell Lung Cancer (NSCLC) Patients under Immune-Checkpoint Inhibitors (ICI)

**DOI:** 10.3390/cancers12102827

**Published:** 2020-09-30

**Authors:** Andrea De Giglio, Laura Mezquita, Edouard Auclin, Félix Blanc-Durand, Mariona Riudavets, Caroline Caramella, Gala Martinez, Jose Carlos Benitez, Patricia Martín-Romano, Lamiae El-Amarti, Lizza Hendriks, Roberto Ferrara, Charles Naltet, Pernelle Lavaud, Anas Gazzah, Julien Adam, David Planchard, Nathalie Chaput, Benjamin Besse

**Affiliations:** 1Cancer Medicine Department, Gustave Roussy, 94805 Villejuif, France; andrea.degiglio@studio.unibo.it (A.D.G.); LMEZQUITA@clinic.cat (L.M.); felix.blanc-durand@gustaveroussy.fr (F.B.-D.); MRiudavets@santpau.cat (M.R.); gala.martinez@juntadeandalucia.es (G.M.); Josecarlos.BENITEZ-MONTANEZ@gustaveroussy.fr (J.C.B.); LAMIAE.EL-AMARTI@gustaveroussy.fr (L.E.-A.); lizza.hendriks@mumc.nl (L.H.); Roberto.Ferrara@istitutotumori.mi.it (R.F.); charles.naltet@gustaveroussy.fr (C.N.); PERNELLE.LAVAUD@gustaveroussy.fr (P.L.); david.planchard@gustaveroussy.fr (D.P.); 2Department of Specialized, Experimental and Diagnostic Medicine, S.Orsola-Malpighi University Hospital, Alma Mater Studiorum University of Bologna, 40126 Bologna, Italy; 3Medical Oncology Department, Hospital Clinic, 08036 Barcelona, Spain; 4Laboratory of Translational Genomics and Targeted therapies in Solid Tumors, IDIBAPS, 08036 Barcelona, Spain; 5Medical and Thoracic Oncology Department, Georges Pompidou Hospital, 75015 Paris, France; edouard.auclin@aphp.fr; 6Radiology Department, Gustave Roussy, 94805 Villejuif, France; caroline.caramella@gustaveroussy.fr; 7Early Drug Development Department, Gustave Roussy, 94805 Villejuif, France; Patricia.MARTIN-ROMANO@gustaveroussy.fr (P.M.-R.); Anas.GAZZAH@gustaveroussy.fr (A.G.); 8Department of pulmonary diseases, GROW- School for Oncology and developmental biology, Maastricht UMC+, 6229 Maastricht, The Netherlands; 9Thoracic Oncology Unit, Medical Oncology Department Fondazione IRCCS Istituto Nazionale dei Tumori di Milano, 20133 Milano, Italy; 10Department of Pathology, Gustave Roussy, 94805 Villejuif, France; julien.adam@gustaveroussy.fr; 11Laboratory of Immunomonitoring in Oncology and CNRS-UMS 3655 and INSERM-US23, Gustave Roussy, 94805 Villejuif, France; chaput-gras.nathalie@gustaveroussy.fr; 12Université Paris-Saclay, Faculté de Pharmacie, 92296 Chatenay-Malabry, France; 13Université Paris-Saclay, Faculté de Médicine, 94276 Le Kremlin Bicêtre, France

**Keywords:** non-small cell lung cancer, immunotherapy, steroids

## Abstract

**Simple Summary:**

Recently, the introduction of immunotherapy radically changed the therapeutic algorithm of non-small-cell lung cancer as an upfront or secondary strategy. Unfortunately, the small amount of patient benefits from immune-checkpoint inhibitors (ICI) and the prognostic role of concomitant treatments are a burning open issue. The use of steroids was associated with poor outcomes during ICI. We investigated the impact of intercurrent steroids, according to clinical indication, which is actually unclear. Interestingly, the use of intercurrent steroids given for cancer-unrelated symptoms has no survival impact on our study cohort.

**Abstract:**

Background: Baseline steroids before ICI have been associated with poor outcomes, particularly when introduced due to cancer symptoms. Methods: Retrospective analysis of advanced NSCLC patients treated with ICI. We collected the use of intercurrent steroids (≥10 mg of prednisone-equivalent) within the first eight weeks of ICI. We correlated steroid use with patient outcomes according to the indications. Results: 413 patients received ICI, 299 were steroids-naïve at baseline. A total of 49 patients received intercurrent steroids (16%), of whom 38 for cancer-related symptoms and 11 for other indications, such as immune-related events. Overall, median (m) progression-free survival (PFS) was 1.9 months (mo.) [95% CI, 1.8-2.4] and overall survival (OS) 10 mo. [95% CI, 8.1–12.9]. Intercurrent steroids under ICI correlated with a shorter PFS/OS (1.3 and 2.3 mo. respectively, both *p* < 0.0001). Intercurrent steroids for cancer-related symptoms correlated with poorest mPFS [1.1 mo.; 95% CI, 0.9–1.5] and mOS [1.9 mo.; 95%CI, 1.5–2.4; *p* < 0.0001)]. No mOS and mPFS differences were found between cancer-unrelated-steroid group and no-steroid group. Steroid use for cancer-related symptoms was an independent prognostic factor for poor PFS [HR 2.64; 95% CI, 1.2–5.6] and OS [HR 4.53; 95% CI, 1.8–11.1], both *p* < 0.0001. Conclusion: Intercurrent steroids during ICI had no detrimental prognostic impact if the indication was unrelated to cancer symptoms.

## 1. Introduction

The clinical development of immune-checkpoint inhibitors (ICI) has changed the paradigm of treatment for patients with non-small cell lung cancer (NSCLC) [1]. However, only a limited part of patients benefits from ICI, rising up the need for affordable biomarkers and different therapeutic strategies. Although evaluating the tumor membrane expression of programmed death-ligand 1 (PD-L1) has become mandatory for newly diagnosed advanced NSCLC [2], the ICI as single-agent can achieve satisfying responses regardless of the PD-L1 status in the pretreated population [1].

In addition to tumor biomarkers, other clinical factors have been associated with ICIs outcomes. These factors can potentially influence ICIs efficacy, such as age, Eastern Cooperative Organization performance status (ECOG PS), and concomitant therapies.

Systemic steroids impair the immune system at multiple levels, ranging from the inhibition of acute inflammation to a long-term immunomodulatory effect, and constitute the cross-sectional backbone of the immunosuppressive therapy [3].

In cancer patients, steroids are largely used for treatment of cancer-related symptoms, such as dyspnea, fatigue, anorexia, and symptomatic brain metastasis [4,5,6]. Other conditions, not strictly related to cancer spread, may also require steroids prescription at the beginning or during the anticancer treatments: Exacerbations of chronic diseases (e.g., chronic obstructive pulmonary disease (COPD), rheumatic diseases, etc.) or toxicities developed under therapies, such as immune-related adverse events (irAEs) [7]. 

Baseline use of ≥ 10 mg of prednisone equivalents was associated with poor outcomes to ICIs in a large retrospective cohort of pretreated advanced NSCLC, independently from ECOG PS, presence of brain metastasis, or smoking status [8]. In a similar retrospective work, the early introduction of steroids showed a detrimental impact on survival outcomes [9]. Based on these, the use of ≥ 10 mg of prednisone equivalents under ICI has been limited and it is still a classical baseline exclusion criterion from clinical trials.

However, a recent retrospective study revealed that the negative impact of baseline steroids was only observed among patients treated for cancer-related symptoms, more likely related to the palliative condition of this population. [10]. To date, the clinical relevance of the initiation of steroids under ICI according to the reason for prescription remains unknown. 

In this study, we aimed to assess the impact on patient outcomes of the intercurrent introduction of steroids (within the first eight weeks) under ICI therapy, and according to the reason of prescription in a large cohort of pretreated advanced NSCLC patients under ICI.

## 2. Results

A total of 413 patients treated with ICIs were enrolled (Appendix A). The median age was 63 years (range 30–92), most of them were male (66%), current or former smokers (91%), with non-squamous histology (76%) and ECOG PS ≤ 1 (77%). PD-L1 status was available for 52% of patients, of which 149 (69%) had ≥ 1% PD-L1 expression and 68 (31%) were negative. All patients received single-agent ICI but 29 (combination with anti-EGFR antibody *n* = 18, immunotherapy investigational drugs *n* = 8, radiotherapy *n* = 3). Baseline characteristics of the overall population are summarized in Table 1. With a median follow-up of 24.4 months [95% CI, 19.5 to 32.4], median PFS and OS were 1.9 months [95% CI, 1.77 to 2.40] and 10 months [95% CI, 8.11 to 12.91], respectively.

### 2.1. Steroids in the Study Population

At baseline, 114 patients (28%) received steroids. In the baseline steroids-naïve group of patients (*N* = 299, 72%), 49 (12%) started them within the first eight weeks of ICI therapy. Among them, 38 patients (9%) received steroids for cancer-related indications and 11 (2%) for non-cancer-related indications. 

Baseline characteristics of the study population according to the intercurrent steroids are summarized in Table 1. The intercurrent steroid population was most commonly associated with ECOG PS2 and more than two metastatic sites. Steroids were given orally (80%), with only 20% intravenously prescribed.

The main indications for cancer-related indications were dyspnea (50%), symptomatic brain metastasis (16%), cancer-related pain (8%) (Appendix A). The main indication for non-cancer-related indications were immune-related adverse events (irAEs) (54.6%) and pneumonia/re-exacerbation of COPD (27.1%) (Appendix A).

The median daily dose of prednisone equivalent was 40 mg for cancer-related symptoms (range 5–225, IQR 17.5–60) and 50 mg for unrelated indications (range 10–90, IQR 20–60) (Appendix A). The median time on treatment was eight weeks for cancer-related (range 0.1–53, IQR 4.3–16.6) and five weeks for unrelated indications (range 0.1–121, IQR 0.7–26.1) (Appendix A).

### 2.2. Steroids and Clinical Outcomes

Steroids at baseline were associated with poor PFS and OS (both *p* < 0.0001), with a median PFS of 1.68 months [95% CI, 1.4 to 2.0], and a median OS of 4.20 months [95% CI, 2.6 to 9.3]. 

Intercurrent steroids were also associated with poor PFS and OS (both *p* < 0.0001), with a median PFS of 1.3 months [95% CI, 0.9 to 1.6], and a median OS of 2.23 months [95% CI, 1.94 to 3.91]. Figure 1 shows the survival outcomes according to the steroid therapy in the study population.

Considering the reason of prescription, in the intercurrent steroids group for cancer-related symptoms, the median OS was 1.9 months [95% CI, 1.5 to 2.4] versus 13.4 months [95% CI, 4.30 to not reached (NR)] in the intercurrent steroid group for non-cancer-related symptoms [95% CI, 4.3 to NR]. In terms of PFS, the median was 1.2 months [95% CI, 0.85 to 1.51] versus 2.3 months [95% CI, 1.22 to not reached (NR)] in intercurrent steroids group for non-cancer-related symptoms (both *p* < 0.0001). (Figure 1). In the steroids-naïve population (*N* = 250, 61%), neither at baseline nor intercurrent, the median PFS was 2.6 months [95% CI, 2.2 to 3.9] and a median OS of 13.8 months. [95% CI, 11.4 to 18], with no differences compared to intercurrent steroids group for non-cancer-related symptoms. 

Figure 2 shows the clinical course of patients receiving intercurrent steroids for cancer-related (Appendix A) and unrelated (Appendix A) indications during the first year of follow-up. 

Intercurrent steroids for cancer-related symptoms was an independent factor for PFS [HR 2.64; 95% CI, 1.24 to 5.61; *p* < 0.0001] and OS [HR 4.53; 95%CI, 1.8 to 11.1); *p* < 0.0001] in the multivariate analysis, including age, gender, smoking status, histology, ECOG PS, and antibiotics at baseline (Table 2). The ECOG PS ≥ 2 and a precedent/concomitant use of antibiotics were independently associated with poor PFS (respectively HR 1.5, (95% CI, 1.1 to 2.2) *p* = 0.01, and 1.3, (95% CI,1 to 1.8), *p* = 0.01) and OS (respectively HR 2.1, (95% CI 1.4 to 3.1), *p* < 0.0001, and 1.6, (95% CI, 1.1 to 2.1) *p* = 0.003) (Table 2 and Appendix A). 

## 3. Discussion

As far as we are aware, this is the first study reporting that intercurrent steroids under ICI have no detrimental impact on clinical outcomes when prescribed for non-cancer-related symptoms. In our work, 12% of the patients were treated with intercurrent steroids and experienced poor outcomes (mPFS 1.3 months; mOS 2.2 months). Nevertheless, this negative impact was mainly associated with the prescription for cancer-related symptoms, suggesting that this can be more likely related to the palliative condition of this population, and not with the drug-effect on their immune system. No differences were observed between intercurrent steroids group for cancer-unrelated symptoms (mPFS 2.2 months; mOS 13.4 months) and the no-steroids group (mPFS 2.6 months; mOS 13.8 months).

Daily dose of ≥10 mg of prednisone equivalent has been historically considered an exclusion criterion for clinical trials, mainly based on the increased rate of infections in other chronic diseases, but not on a specific contraindication for cancer patients [11,12,13]. In advanced NSCLC patients, we reported that ≥10 mg use of daily prednisone equivalent at baseline was strongly associated with poor ICI outcomes in a large cohort of 640 patients [8]. However, patients treated with steroids were more commonly associated with poor prognostic factors, such as brain metastasis or ECOG PS ≥ 2, that may also play a role in the negative impact reported. 

Based on this hypothesis, Ricciuti et al. studied the clinical impact of baseline-steroids, with the same threshold, according to the reason of prescription in 650 advanced NSCLC patients [10]. In this work, the authors observed that the negative impact of baseline-steroids on outcomes was not confirmed for the subgroup treated for cancer-unrelated indications, questioning the negative predictive role suggested for steroids on ICI treatment [10], which is in line with our data. 

Although some previous studies have also investigated the role of intercurrent steroids treatment. Fucà et al. [9] reported worse outcomes in 23% of patients treated with intercurrent steroids (first 28 days) retrospectively studied on 151 advanced NSCLC patients treated with ICI. Similarly, Drakaki et al. [14] reported that baseline-steroids and intercurrent steroids up to 30 days under ICI (30%) were associated with poor outcomes in a cohort of 862 NSCLC patients. In these two studies, the reason for prescription was not analyzed and the threshold for daily prednisone equivalents differed. Interestingly, Pennell et al. reported at the 2019 World Conference on Lung Cancer (WCLC) the impact of intercurrent steroids on the real-world CancerLinq Discovery Database in 11,143 advanced NSCLC treated with ICI [15]. Among them, 1581 received intercurrent steroids (≥10 mg of prednisone equivalent) within the first 30 days, associated with poor OS. However, when the survival was adjusted by clinical characteristics (age/gender/PS/brain metastasis) no differences were observed in OS, consistently with our findings. The lack of detailed clinical data about the reason for prescription and the appurtenance of a poor prognosis subgroup may confound this statement. In fact, only one experience evidenced this conflict dissecting the clinical indication as palliative or not palliative [10]. 

Recently, a meta-analysis explored the impact of steroids across 16 studies with melanoma and NSCLC patients [16]. Overall, the patients treated with steroids had a higher risk of both progression and death, but only the intercurrent steroids for palliative purposes were related to the greater risk of death compared to patients receiving steroids for brain metastasis or irAEs. However, the clinical indication was not formally investigated in the majority of the studies enrolled in this meta-analysis. Conversely, the use of steroids for the management of irAEs did not increase the risk of death, as previously described. 

The main biological hypothesis of the detrimental impact of steroids under ICI is based on the well-known immunosuppressive effects on both acute and chronic inflammation. The activation of glucocorticoid intracellular receptor (GR) inhibits the innate and adaptive immune response via genomic and non-genomic pathways [2]. Nevertheless, an increasing amount of evidence also shows that steroids can positively induce the innate response stimulating the expression of pattern recognition receptors (PRR), complement components [17], and the production of several cytokines and their receptors such as IL-1, IL-6, TNF, and IFNγ [18]. In light of these data, Cain et al. [17] proposed a biphasic model of immune systems’ regulation under the stimulus of steroids that may promote an immune response earlier but shorter with respect to a condition of endogenous steroids deficiency. Whether this model is applicable in cancer patients treated concomitantly with ICI and steroids remains an open issue but evocates possible different biological scenarios, in particular, involving the innate system and myeloid lines.

Consistently, as previously reported, steroids can induce increased absolute neutrophil count (ANC) [19,20,21], and it has been even associated with the polarization of macrophages to M2 macrophages, with pro-tumoral functions [22]. Thus, the global impact of steroids on the immune context of the patients should be further explored. 

Although it was not the primary goal of the project, our study showed that the introduction of antibiotic therapy within two months before and one month after the ICI was independently associated with poor outcomes at the multivariate analysis. Interestingly, antibiotics may alter gut microbiota, as suggested by pre-clinical studies conducted on mice models [23]. Moreover, several retrospective analyses [24,25] confirmed the detrimental role of antibiotics in survival outcomes in cohorts of advanced NSCLCs treated with anti-PD-(L)1 inhibitors, albeit the assessment of the underlying mechanisms is still an open issue.

Finally, our findings also confirmed that a deteriorated performance status at baseline (≥2) correlated with poor survival outcomes, as previously described [26]. Notably, Facchinetti et al. evidenced that only a disease burden-induced poor ECOG PS (vs. comorbidities-induced) was independently related with a bad prognosis [27]. 

Our study has several limitations worthy of discussion. Firstly, the analysis had the intrinsic limitation of a monocentric retrospective experience, with some missing data (e.g., PD-L1 status). In addition, response assessment was not homogenously performed following RECIST v.1.1 criteria with a dedicated radiologist, but the response rate was investigator-assessed, thus resulting less accurate for PFS analysis, the reason why we considered it as secondary endpoint. Finally, the small size of the sample of patients treated for non-cancer-related indications did not allow postulating definitive conclusions.

Despite these limitations, we believe that our work helps to lighten the role of intercurrent steroids in patients treated with ICI. In particular, that the negative impact on clinical outcomes of intercurrent-steroids, as at baseline, is more likely related to the palliative condition of the patients rather than a drug-effect. Thus, steroid prescription should not be avoided under ICI if it is indicated.

## 4. Materials and Methods 

### 4.1. Patients 

We conducted a retrospective study of advanced NSCLC patients treated with PD-(L) 1 ICI, as monotherapy or in combination with other drugs (e.g., ipilimumab), at Gustave Roussy (GR), between February 2013 and September 2018. Patients treated with a combination of chemotherapy and ICI were not included in this study. 

Clinical and biological data were collected from electronic medical records. PD-L1 expression was assessed by IHC on tumor cells, according to standard practice. PD-L1 expression ≥1% was considered positive. This study was approved by the Internal Review Board at GR. Steroids at baseline and/or within the first eight weeks of ICI (= intercurrent steroids) were retrospectively collected, including type of drug, daily dose, duration of therapy, way of administration, and reason for prescription. We defined the use of steroid therapy if the was at least one-day prescription of ≥ 10 mg of prednisone-equivalent.

### 4.2. Statistical Analysis

Median (interquartile-range) values and proportions (percentage) were provided for the description of continuous and categorical variables, respectively. Mean and proportions were compared using t-test (or Anova if appropriate) and chi2-test (or Fisher’s exact test, if appropriate), respectively. The primary outcome was overall survival (OS), secondary end-point was progression-free survival (PFS). OS was defined as the time between the start of ICI treatment and death. PFS was defined as the time from the start of ICI treatment to the first radiological or clinical progression according to RECIST 1.1 or per investigator discretion, whichever occurred first. Patients who were still alive at data lock (May 2019) were censored at last contact. Median OS and PFS were estimated with the Kaplan Meier method and compared with the Log Rank test. The median follow-up was calculated with the reverse Kaplan Meier method.

The association between clinical, biological, pharmacological variables and survival endpoints was explored with a Cox model regression. First, a univariate analysis was performed for both survival endpoints. All variables reaching a *p* value < 0.1 or considered as clinically relevant were included in a multivariable model. *p* value < 0.05 were considered statistically significant. Statistical analyses were performed using R-Studio.

## 5. Conclusions

Our data suggest that the intercurrent steroids of a daily dose of ≥10 mg of prednisone equivalent within the first eight weeks of ICI have no detrimental impact on prognosis if the indication is not related to cancer symptoms. A larger prospective validation of our retrospective data is needed.

## Figures and Tables

**Figure 1 cancers-12-02827-f001:**
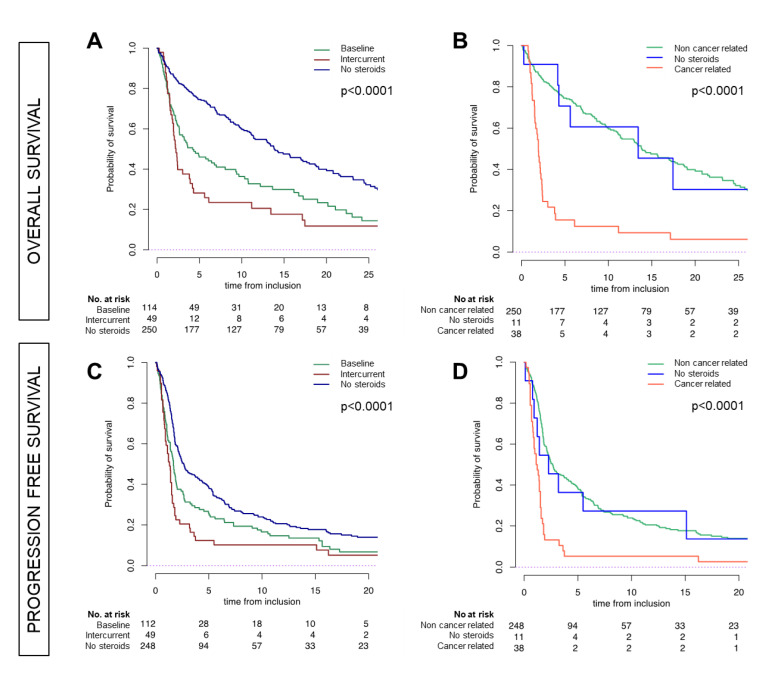
Outcomes to ICI treatment in the group of patients treated with intercurrent steroids for tumor-related indication compared with the group of patients receiving intercurrent steroids for tumor-unrelated indications and with the group of steroid-free patients. (**A**) shows overall survival (OS) in baseline-treated patients with steroids (green line), steroids free population (blue line), and intercurrent steroids treated patients (red line). (**B**) shows OS in steroid-free population (blue line), intercurrent steroid treatment for cancer-related indications (orange line), and intercurrent steroids treatment for non-cancer-related indications (green line) (cancer-related vs. non-cancer-related, *p* < 0.002). (**C**) As in (**A**) but progression-free survival (PFS), (**D**) as in (**B**) but PFS (cancer-related vs. non-cancer-related *p* = 0.05).

**Figure 2 cancers-12-02827-f002:**
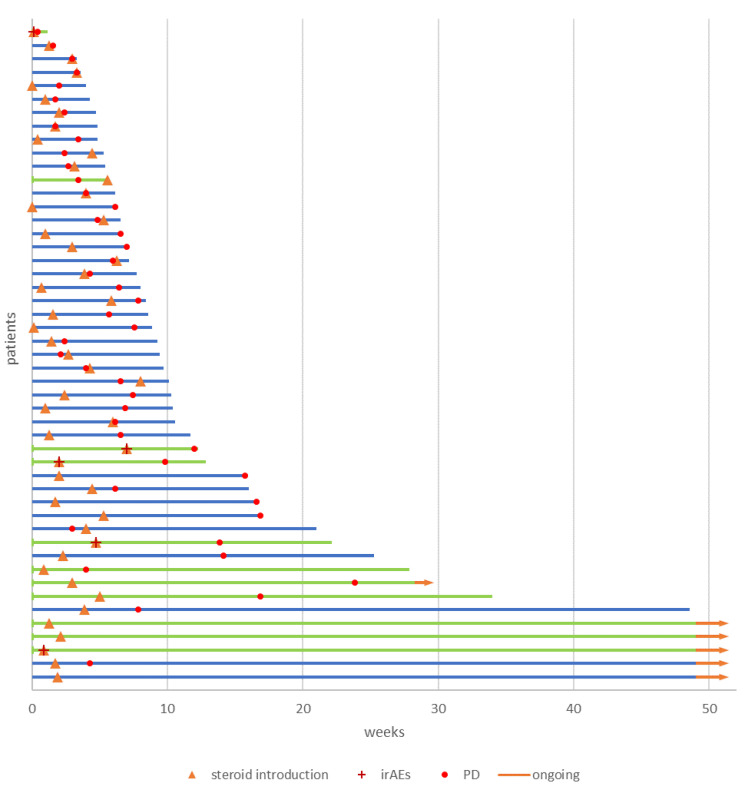
Swimmers plot detailing progression survival (PFS) and overall survival (OS) after one year of follow-up of patients receiving intercurrent steroids for cancer-related (blue) and not related (green) indications. Abbreviations: Immune-related adverse events (irAEs); progressive disease (PD).

**Table 1 cancers-12-02827-t001:** Baseline characteristics of the study population.

Characteristic	Subgroup	Overall	Steroid-Free Population	Baseline Steroids	Intercurrent Steroids Non-Cancer-Related	Intercurrent Steroids Cancer-Related	*p* Value
		(*N* = 413)	(*N* = 250)	(*N* = 114)	(*N* = 11)	(*N* = 38)	
**Age** **Median, range**		63 (30;92)	63 (30;92)	62 (34;85)	68 (57;75)	63 (42;80)	0.38
**Gender**	F	140 (34%)	88 (35%)	33 (29%)	6 (55%)	13 (34%)	0.41
	M	273 (66%)	162 (66%)	81 (71%)	5 (45%)	25 (66%)	
**Smoking history**	current	127 (31%)	81 (33%)	31 (27%)	4 (36%)	11 (30%)	0.95
	former	239 (59%)	139 (57%)	74 (66%)	6 (55%)	20 (55%)	
	never smoker	39 (9%)	25 (10%)	8 (7%)	1 (9%)	5 (14%)	
	missing	8	5	1	0	2	
**Histology**	Non squamous	314 (76%)	195 (78%)	82 (72%)	8 (72.7%)	29 (76%)	0.83
	Squamous	99 (24%)	55 (22%)	32 (28%)	3 (27.2%)	9 (24%)	
**PD-L1**	negative	67 (32%)	42 (32%)	18(31%)	2 (25%)	5 (36%)	0.93
	positive ≥ 1%	145 (68%)	89 (68%)	41 (69%)	6 (75%)	9 (64%)	
	missing	201	119	55	3	24	
**Stage**	IIIB	3 (0.7%)	3 (1.2%)	0	0	0	0.59
	IV	410 (99.3%)	247 (98.8%)	114	11	38	
**Metastatic sites**	≤2	188 (46%)	123 (49%)	47(41%)	5 (45.4%)	13 (34%)	0.21
	>2	224 (54%)	126 (51%)	67 (59%)	6 (54.5%)	25 (66%)	
	missing	1	1	1	0	0	
**Brain metastases**	No	286(69%)	192 (77%)	56 (49%)	6 (55%)	32 (84%)	0.12
	Yes	127 (31%)	58 (23%)	58 (51%)	5 (45%)	6 (16%)	
**Liver metastases**	No	311 (75%)	188 (75%)	86 (75%)	10 (90%)	27 (71%)	0.44
	Yes	102 (25%)	62 (25%)	28 (25%)	1 (10%)	11 (29%)	
**Line of immunotherapy**	1–2	259 (63%)	167 (67%)	58 (51%)	6 (55%)	28 (74%)	0.46
	>2	154 (37%)	83 (33%)	56 (49%)	5 (45%)	10 (26%)	
**ECOG PS**	0–1	313 (77%)	207 (84%)	72 (64%)	8 (72.7%)	26 (70%)	0.09
	≥2	95 (23%)	40 (16%)	41(36%)	3 (27.2%)	11 (30%)	
	missing	10	3	5	0	1	

Abbreviations: ECOG PS, Eastern Cooperative Oncology Group performance status.

**Table 2 cancers-12-02827-t002:** Multivariate analysis for PFS and OS.

Multivariate Analysis	PFS			OS		
	HR	95% CI	*p* Value	HR	95% CI	*p* Value
**Age**						
**≤65 years**	1	-		1	-	
**>65 years**	0.97	0.74–1.27	0.85	1.1	0.80–1.50	0.53
**Smoking status**						
**Never/former smoker**	1	-		1	-	
**Smoker**	0.6	0.40–0.89	0.01	0.88	0.56–1.37	0.58
**Histology**						
**Non-squamous**	1	-		1	-	
**Squamous**	1.11	0.81–1.53	0.48	1.25	0.86–1.80	0.22
**Immunotherapy line**						
**≤2**	1	-		1	-	
**>2**	1.28	0.97–1.68	0.07	1.23	0.89–1.68	0.19
**Number of met. Sites**						
**≤2**	1	-		1	-	
**>2**	1.15	0.87–1.51	0.31	1.26	0.92–1.72	0.14
**ECOG PS**						
**0–1**	1	-		1		
**≥2**	1.56	1.10–2.21	0.01	2.15	1.46–3.1	<0.0001
**Use of antibiotics**						
**No**	1					
**Yes**	1.39	1.05–1.83	0.01	1.59	1.17–2.17	0.003
**Intercurrent steroids introduction**						
**Non-cancer-related**	1	-		1	-	
**No steroids**	1.02	0.51–2.02		1.25	0.54–2.88	
**Cancer-related**	2.64	1.24–5.61	<0.0001	4.53	1.84–11.12	<0.0001

Legend: Cox proportional hazard regression analysis of progression-free survival and overall survival according to age, smoking status, histology, immunotherapy line of treatment, number of metastatic sites, ECOG PS, use of antibiotics and intercurrent introduction of corticosteroids for cancer-related symptoms. Abbreviations: ECOG PS, Eastern Cooperative Oncology Group performance status; HR, hazard ratio; CI, confidence interval.

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
