# Peer review of "Impact of Intercurrent Introduction of Steroids on Clinical Outcomes in Advanced Non-Small-Cell Lung Cancer (NSCLC) Patients under Immune-Checkpoint Inhibitors (ICI)"

_cancers, 2020, doi:10.3390/cancers12102827_

Round 1

Reviewer 1 Report

To the authors,

The authors have provided a retrospective overview of 413 patients treated at a single institute with immunotherapy for advanced NSCLC. Subgroup analysis comparing patients whom have had received prednisone >=10mg (or equivalent) on a daily basis during the initial 8 weeks after start of therapy versus non-steroid use or those whom had it at baseline. Comparison of PFS and OS shows a significant higher OS/PFS in patients treated with prednisone for non-cancer related indications (mostly adverse events on ICI) compared to cancer related indications (brain mets, dyspnea, fatigue). Overall this is a well written manuscript with a clear message. There are several limitations, mainly the size of the subgroup of patients receiving steroids for non-cancer related indications. However the authors have addressed these already in their discussion.

Points of discussion:

  • In the current manuscript it is unclear if stage may have had an effect on prognosis as these data are not provided (i.e. stage III unresectable disease (and no RTx), stage IV a-b-c etc)
  • Line 244-245 “We conducted a retrospective study of advanced NSCLC patients treated with PD-(L)1 ICI, as monotherapy or in combination with other drugs”. What combinations of drugs were assessed and could this have had an influence on the results?
  • The rate of corticosteroid use is relative high (24%) compared to other studies (also referenced in this manuscript). Is this due to the relatively higher frequency of >2 lines of immunotherapy (centre selection)? (also in the corticosteroid group compared to the non-corticosteroid group). Could this have affected outcome in the non-cancer related group and what where the reasons for corticosteroid treatment in the at baseline cohort? Was this corticosteroid treatment at baseline initiated at the study centre or at an referral centre? Addition of these data could give more insight on for example indication of corticosteroid for irAE’s in earlier treatment lines as this was the most important reason for corticosteroids in the non-cancer related indication group.
  • Table 2 PFS smoking status, p value 0.01  does not match the CI (typo?).

Reviewer 2 Report

This is a very well designed, conducted and presented mono centric, retrospective analysis of the prognostic impact of intercurrent steroids on clinical outcomes in patients with advanced NSCLC according to the indication of the prescription of steroids. The main strength of the manuscript is that it differentiates the indication of steroid use to cancer-related and cancer-unrelated (eg irEs) symptoms. Minor comments:

  1. The cluster of patients who received steroids for cancer-unrelated indications is very small (N=11) and does not allow for safe conclusions to be drawn. This should be clearly stated in the discussion session. The conclusion that "steroid use does not affect outcome if indication is unrelated to cancer symptoms" is based on data from only 11 patients and should be interpreted with caution
  2. It is not clear from table 1 whether patients received IO as mono therapy or together with chemotherapy.Please add data on table 1 because chemo-IO combo may affect steroid impact on clinical outcomes
  3. There are minor incomprehensible sentences in some points of the manuscript (ie lines 201, 217, etc). Please rephrase

Reviewer 3 Report

Comments for authors:

The manuscript entitled “Impact of intercurrent introduction of steroids on clinical outcomes in advanced non-small-cell lung cancer (NSCLC) patients under immune-checkpoint inhibitors (ICI)” by De Giglio et al.” addresses the impact of intercurrent steroids given for cancer unrelated vs cancer related. Albeit the study delivers interesting results, some critical points should be improved.

Major concerns:

  1. The aim of the study is to demonstrate that the intercurrent steroids given for cancer unrelated has a different impact that those given for immune-related or other symptoms. However, the analysis also includes a cohort of steroid naive patients, apparently as a reference group. To better understand the differences between the three groups, all categories for each variable presented in the multivariate analysis (Table 2) should be included. Authors should present all categories for each variable, for example No steroids, steroids for cancer-related, and steroids for non-cancer related, each one with their associated HR, including HR=1 in the category considered as reference.
  2. When analyzing the Kaplan meier for OS and PFS in figure 1. Authors report 1.9 vs 13.4 median months in OS in steroids for cancer-related vs non-cancer related symptoms. Moreover, the lines in the graph in Figure 1B seem pretty distinct.” The authors need to report the p-value of the analysis of the mPFS and mOS of these two groups. As observed in figure 2, unrelated steroids treated patients tend to have longer survival.
  3. At the same time, the graph also includes the no-steroids group and a p-value. Do this p-value correspond to the Kaplan meier of the three categories?

Strength: The authors suggest that the negative impact in the survival was mainly associated with the prescription for cancer-related symptoms, suggesting that this can be more likely related to the palliative condition of this population. This is a good comment. Even considering that the dose and the duration of the treatment was not different between the two groups.

Minor concerns:

 Improve Figure 1: Change colors or make lines thicker so groups are identified more easily.

  1. Report abbreviation for PS2 in the text. Or better report ECOG PS2.
  2. Check text for some language mistakes (line 183 – have been also investigated????) Line 201 studies (studied). 207 increasing evidence, 217 association. 234 was investigators?? 236 analyzing
  3. In the same way authors describe that the palliative condition rather than drug-effect of steroids might be the cause of lower survival, this may also be the reason why the antibody treatment is associated with poorer outcome. Was the antibiotic treatment or the infection the cause of poor outcome? This idea should be introduced.
